# Quantum walks on graphs of the ordered Hamming scheme and spin networks

**Hiroshi Miki [1⋆], Satoshi Tsujimoto[2†] and Luc Vinet[3‡]**

**1** Meteorological College, Asahi-Cho, Kashiwa 277 0852, Japan
**2** Department of Applied Mathematics and Physics, Graduate School of Informatics,
Kyoto University, Sakyo-Ku, Kyoto 606 8501, Japan
**3** Centre de Recherches Mathématiques, Université de Montréal,
PO Box 6128, Centre-ville Station, Montréal (Québec), H3C 3J7, Canada

⋆ hmiki@mc-jma.go.jp, † tsujimoto.satoshi.5s@kyoto-u.jp, ‡ vinet@crm.umontreal.ca

## Abstract

**It is shown that the hopping of a single excitation on certain triangular spin lattices with non-uniform couplings and local magnetic fields can be described as the projections of quantum walks on graphs of the ordered Hamming scheme of depth 2. For some values of the parameters the models exhibit perfect state transfer between two summits of the lattice. Fractional revival is also observed in some instances. The bivariate Krawtchouk polynomials of the Tratnik type that form the eigenvalue matrices of the ordered Hamming scheme of depth 2 give the overlaps between the energy eigenstates and the occupational basis vectors.**



# 1 Introduction

This paper introduces two-dimensional spin lattices that exhibit perfect state transfer between two single locations and multi-site fractional revival on a one-dimensional subset of the lattice. These novel models are obtained by projecting quantum walks on graphs that belong to the ordered Hamming scheme which generalizes the well known Hamming association scheme. On the one hand, continuous walks on graphs have been used to formulate various computation algorithms [9, 10, 13]. On the other hand, the 1-excitation dynamics of spin chains has attracted attention as a mean to realize the transport of quantum states with a minimum of external controls [6, 7, 21, 27]. One speaks of perfect state transfer (PST) when the transport between two locations happens with probability one. It has been appreciated that non-uniform couplings and possibly local magnetic fields are required to achieve PST over distances of more than three sites [11]. One analytic model that admits PST over (reasonably) arbitrary distances has couplings given by the Krawtchouk polynomials recurrence coefficients [1]. Interestingly, it has been observed [11] that the 1-excitation dynamics of this Krawtchouk chain can actually be obtained by projecting quantum walks on the 1-link hypercube to a weighted path. It will be recalled that the hypercube is one of the simplest graphs of the Hamming scheme [8, 29]. The fact that the Krawtchouk polynomials arise naturally in that scheme is not foreign to the connection we just mentioned. The end-to-end PST in the chain can thus be seen as a manifestation of the fact that there is also PST between antipodal points of the hypercube. These results have motivated extensive examinations of quantum walks and especially of PST on graphs [17, 22].

The coherent transport of states on higher dimensional spin lattices has also been explored. A few models with interesting transfer properties [24, 25, 28] have been designed using the theory of multivariate Krawtchouk polynomials [12, 18, 19, 30]. These systems exhibit fractional revival (FR) whereby an initially localized state is reproduced periodically in a number of fixed locations [2, 16]. In view of the relation between the hypercube and the Krawtchouk chain, it is natural to enquire if such models could not be obtained from the projection of quantum walks in higher dimensional graphs. In pursuing that question we will in fact identify graphs in a generalization of the Hamming scheme with dynamics that projects to 1-excitation hopping on a triangular lattice exhibiting perfect state transfer and multi-site fractional revival. We suggest that these systems could be realized as photonic lattices and possibly be of use for certain algorithms.

The paper will be organized as follows. The definition of the ordered Hamming scheme of depth 2 will be recalled in section 2. Particular graphs in that scheme will be identified in section 3 and the dynamics governed by their adjacency matrices will be shown to project to 1-excitation Hamiltonians for a triangular lattice of spins in the plane. The bivariate Krawtchouk polynomials of the Tratnik type will be introduced in section 4 to obtain the energy eigenstates. The transport properties will be examined in section 5 and it will be found that there is perfect state transfer between two specific summits of the triangular lattice. The paper will end with concluding remarks.

# 2 The ordered Hamming scheme of depth 2

Let $Q = \mathbb{Z}/2\mathbb{Z}$. Consider the set $Q^{(n,r)}$ of vectors of dimension $nr$ over $Q$. The vector $x \in Q^{(n,r)}$ will be presented by the $r$-binary sequences of length $n$ over $Q$:

$$x = (\bar{x}_1, \bar{x}_2, \ldots, \bar{x}_n),$$

where $\overline{x}_j = (x_{j1}, x_{j2}, \ldots, x_{jr}) \in Q^r$. We define the shape $e$ of $x \in Q^{(n,r)}$ by

$$e(x) = (e_1, e_2, \cdots e_r),$$
$$e_i = \#\{j \in \{1, 2, \cdots, n\} \mid x_{ji} = 1, x_{j,i+1} = x_{j,i+2} = \cdots = x_{jr} = 0\}$$

and denote the set of the all shapes by

$$E = e(Q^{(n,r)}) = \{(e_1, e_2, \cdots, e_r) \in (\mathbb{Z}_{\geq 0})^n \mid 0 \leq e_1 + e_2 + \cdots + e_r \leq n\}.$$

For example, $a = (00, 10, 11, 00, 01) \in Q^{(5,2)}$ is a 2-binary sequence of length 5 and $e(a) = (1, 2)$. For two vectors $x, y \in Q^{(n,r)}$, we shall write $x \sim_e y$ if the shape of $x - y$ is equal to $e$. Then we can introduce the graph $G_e$ associated with the shape $e$ as the one where all two vertices $(v_x, v_y)$ in $\{v_x \mid x \in Q^{(n,r)}\}$ are linked if $v_x \sim_e v_y$; the corresponding adjacency matrix is given by

$$A_e = (a_{x,y}), \quad a_{x,y} = \begin{cases} 1 & (x \sim_e y) \\ 0 & (\text{otherwise}). \end{cases}$$

It is known that $\mathcal{A} = \{A_e \mid e \in E\}$ forms an association scheme. It is called the ordered Hamming scheme of depth $r$ [5, 23].

In this paper, for fixed positive integer $N$, we shall consider the ordered Hamming scheme of depth 2 $(Q^{(N,2)}, \mathcal{A})$ where the set of adjacency matrices

$$A_{(i,j)} \quad 0 \leq i + j \leq N$$

form the (commutative) Bose-Mesner algebra:

$$A_{(i,j)}A_{(k,l)} = \sum_{0 \leq i'+j' \leq N} \alpha^{(i',j')}_{(i,j),(k,l)} A_{(i',j')}.$$

The intersection numbers $\alpha^{(i',j')}_{(i,j),(k,l)}$ are equal to the number of vertices $z$ such that $e(x-z) = (i, j)$ and $e(y-z) = (k, l)$ if $e(x-y) = (i', j')$ for $x, y, z \in Q^{(N,2)}$. In particular, one has the following explicit formulas involving $A_{(1,0)}$ and $A_{(0,1)}$.

$$A_{(1,0)}A_{(i,j)} = (N+1-i-j)A_{(i-1,j)} + jA_{(i,j)} + (i+1)A_{(i+1,j)} \tag{1}$$
$$A_{(0,1)}A_{(i,j)} = 2(N+1-i-j)A_{(i,j-1)} + 2(i+1)A_{(i+1,j-1)} \tag{2}$$
$$+ (j+1)A_{(i-1,j+1)} + (j+1)A_{(i,j+1)}.$$

It is not difficult to obtain the above relations. Let us write

$$A_{(1,0)}A_{(i,j)} = a_{i,j}A_{(i-1,j)} + b_{i,j}A_{(i,j)} + c_{i,j}A_{(i+1,j)}.$$

The coefficient $a_{i,j}$ stands for how many $z$ exist such that $e(x-z) = (1, 0)$ and $e(y-z) = (i, j)$ if $e(x-y) = (i-1, j)$. For example, take $y = (00, 00, \cdots, 00)$ and $x = (10, 10, \cdots, 10, 01, 01, \cdots, 01, 00, 00, \cdots, 00)$ with $i-1$ 10s, $j$ 01s (11s or the combination of 01 and 11 is also possible) and $N+1-i-j$ 00s. In this situation, $z$ must be obtained by changing one of the $N+1-i-j$ 00s by 10 and there are $N+1-i-j$ ways of doing this. It is easy to see that the number of ways does not depend on the choice of the elements $y \in Q^{(N,2)}$. We thus have $a_{i,j} = N+1-i-j$. The number $c_{i,j}$ of $z$ such that $e(x-z) = (1, 0)$ and $e(y-z) = (i, j)$ if $e(x-y) = (i+1, j)$ can similarly be obtained. Take $y = (00, 00, \cdots, 00)$ and $x$ consisting of $i+1$ 10s and $j$ 01s and $N-1-i-j$ 00s. We see that a $z$ can be obtained by changing one of the $i+1$ 10s by 00s and there are $i+1$ ways of doing that. It thus follows that $c_{i,j} = i+1$. For $b_{i,j}$, take $y = (00, 00, \cdots, 00)$ and $x = (10, 10, \cdots, 10, 01, 01, \cdots, 01, 00, 00, \cdots, 00)$ with $i$ 10s, $j$ 01s and $N-i-j$ 00s. Clearly a $z$ such that $e(x-z) = (1, 0)$ and $e(y-z) = (i, j)$ if $e(x-y) = (i, j)$

can be obtained by changing one of the $j$ 01s by 11 and there are $j$ possible choices. As a result $b_{i,j} = j$ and (1) holds.

Formula (2) is derived in the same fashion. We shall therefore only indicate how the factor in front of $A_{(i,j-1)}$ is identified. This coefficient stands for how many $z$ there are such that $e(x - z) = (0, 1)$ and $e(y - z) = (i, j)$ if $e(x - y) = (i, j - 1)$. Take $y = (00, 00, \cdots, 00)$ and $x = (10, 10, \cdots, 10, 01, 01, \cdots, 01, 00, 00, \cdots, 00)$ with $i$ 10s, $j - 1$ 01s and $N + 1 - i - j$ 00s. To obtain such a $z$, we must change one of the $N + 1 - i - j$ 00s by 01 or 11 and there are $2(N + 1 - i - j)$ ways of doing this.

## 3  Special weighted graphs and their projections

Let us consider the graph $G_{\alpha,\beta}$ whose adjacency matrix is $\alpha A_{(1,0)} + \beta A_{(0,1)}$ with $\alpha, \beta \in \mathbb{R}_{\geq 0}$. We shall call this graph ordered Hamming graph (of depth 2). Now, following [4,11], we consider the projection of the quantum walk on the ordered Hamming graph $G_{\alpha,\beta}$ to the "column subspaces" to identify the corresponding spin lattice.

To the vertices $x \in V = Q^{(N,2)}$ ($|V| = (2^2)^N = 4^N$), we associate orthonormalized vectors $|x\rangle$ such that

$$\langle x \mid y \rangle = \begin{cases} 1 & (x \sim_{(0,0)} y) \\ 0 & (\text{otherwise}) \end{cases}$$

for $x, y \in V$. In this notation the entries $A_{xy}$ of $A$ can be written as $\langle x|A|y\rangle$. Let $(0) \equiv (00, 00, \cdots, 00)$ denote a corner and organize $V$ as a set of $\binom{N+2}{2}$ columns $V_{i,j}$ $(0 \leq i + j \leq N)$ defined by

$$V_{i,j} = \{x \in V, \quad e(x) = (i, j)\}.$$

The number $k_{i,j}$ of vertices in the column $V_{i,j}$ is given by

$$k_{i,j} = \binom{N}{i, j} 2^j,$$

where $\binom{N}{i,j} = \frac{N!}{i!j!(N-i-j)!}$ is the trinomial coefficient. The number $k_{i,j}$ can be identified by taking it into account that vertices with shape $(i, j)$ consist of $i$ 10s, $j$ 01 or 11s and $N - i - j$ 00s. Let us then label the vertices in column $V_{i,j}$ by $V_{(i,j),k}, k = 1, \ldots k_{i,j}$. Under the relation defined by the shape $(1, 0)$, each $V_{(i,j),k}$ in the column $V_{i,j}$ is connected to $N - i - j$ elements of column $V_{i+1,j}$. Similarly, according to the association generated by the shape $(0, 1)$, each $V_{(i,j),k}$ in $V_{i,j}$ is linked with $2(N - i - j)$ elements of column $V_{i,j+1}$ and $j$ elements of column $V_{i+1,j-1}$. Furthermore, with respect to $(1, 0)$ each $V_{(i,j),k}$ in $V_{i,j}$ is connected to $j$ elements of the same column $V_{(i,j)}$.

Consider now the column space taken to be the linear span of the column vectors given by

$$|\text{col } i, j\rangle = \frac{1}{\sqrt{k_{i,j}}} \sum_{k=1}^{k_{i,j}} |V_{(i,j),k}\rangle, \quad 0 \leq i + j \leq N.$$

Since every vertex in column $V_{i,j}$ is connected to the same number of vertices in columns $V_{i+1,j}$, $V_{i,j+1}$ and $V_{i+1,j-1}$ with respect to $(1, 0)$ and $(0, 1)$, $A_{(1,0)}$ and $A_{(0,1)}$ preserve the column space.

Let us compute the matrix elements of $A_{(1,0)}$ and $A_{(0,1)}$ in the basis of the column subspace.

$$\langle \text{col } i+1, j | A_{(1,0)} | \text{col } i, j \rangle = \frac{1}{\sqrt{k_{i+1,j}k_{i,j}}} \sum_{l=1}^{k_{i+1,j}} \sum_{k=1}^{k_{i,j}} \langle V_{(i+1,j),l} | A_{(1,0)} | V_{(i,j),k} \rangle$$

$$= \frac{1}{\sqrt{k_{i+1,j}k_{i,j}}}(N-i-j)k_{i,j}$$

$$= \sqrt{(i+1)(N-i-j)}.$$

To derive the second relation, one can first pick a vertex in $V_{i,j}$, compute the scalar products with the $(N-i-j)$ vertices to which it is linked in $V_{i+1,j}$ and then sum over the $k_{i,j}$ vertices in $V_{i,j}$. Similary, we also have

$$\langle \text{col } i, j | A_{(1,0)} | \text{col } i, j \rangle = j,$$

$$\langle \text{col } i, j+1 | A_{(0,1)} | \text{col } i, j \rangle = \sqrt{2(j+1)(N-i-j)},$$

$$\langle \text{col } i+1, j-1 | A_{(0,1)} | \text{col } i, j \rangle = \sqrt{2(i+1)j}.$$

To conclude, the quantum walk on the ordered Hamming graph $G_{\alpha,\beta}$ is equivalent to the one-excitation dynamics of the spin lattice of triangular shape governed by the following Hamiltonian:

$$H = \sum_{0 \leq i+j \leq N} \alpha \sqrt{(i+1)(N-i-j)} \frac{\sigma_{i,j}^x \sigma_{i+1,j}^x + \sigma_{i,j}^y \sigma_{i+1,j}^y}{2}$$

$$+ \beta \sqrt{2(j+1)(N-i-j)} \frac{\sigma_{i,j}^x \sigma_{i,j+1}^x + \sigma_{i,j}^y \sigma_{i,j+1}^y}{2} \quad (3)$$

$$+ \beta \sqrt{2(i+1)j} \frac{\sigma_{i,j}^x \sigma_{i+1,j-1}^x + \sigma_{i,j}^y \sigma_{i-1,j+1}^y}{2} + \alpha j \frac{1 + \sigma_{i,j}^z}{2}.$$

The lattice sites are labelled by two integers $i, j$ between 0 and $N$ and such that their sum is smaller or equal to $N$. Indeed, on the subspace spanned by the 1-excitation orthonormal basis vectors $|e_{i,j}\rangle$ $(0 \leq i+j \leq N)$, we see that

$$H |e_{i,j}\rangle = \alpha \sqrt{(i+1)(N-i-j)} |e_{i+1,j}\rangle + \beta \sqrt{2(j+1)(N-i-j)} |e_{i,j+1}\rangle$$

$$+ \alpha j |e_{i,j}\rangle + \alpha \sqrt{i(N+1-i-j)} |e_{i-1,j}\rangle + \beta \sqrt{2j(N+1-i-j)} |e_{i,j-1}\rangle \quad (4)$$

$$+ \beta \sqrt{2(i+1)j} |e_{i+1,j-1}\rangle + \beta \sqrt{2i(j+1)} |e_{i-1,j+1}\rangle,$$

thus confirming the equivalence.

## 4 Bivariate Krawtchouk polynomials and energy eigenstates

In the Hamming scheme, the univariate Krawtchouk polynomials

$$K_n^N(x; p) = {}_2F_1 \left( \begin{matrix} -n, -x \\ -N \end{matrix}; \frac{1}{p} \right) = \sum_{l=0}^{\infty} \frac{(-n)_l(-x)_l}{l!(-N)_l} \left( \frac{1}{p} \right)^l, \quad x, n = 0, 1, \cdots, N$$

come up and are applied to analyze the properties of the quantum walks on the associated graphs [11], where $(a)_n = a(a+1)\cdots(a+n-1)$ is the standard Pochhammer symbol. In the ordered Hamming scheme of depth 2, the bivariate Krawtchouk polynomials of the Tratnik

type appear, as pointed out in the related coding theory [5]. These two-variable orthogonal polynomials are defined as the following product of the univariate Krawtchouk polynomials:

$$T_{m,n}^N(x,y) = \frac{1}{(-N)_{m+n}} k_m^{N-n}(x;p) k_n^{N-x}(y;\frac{q}{1-p}), \quad 0 \le x+y, m+n \le N,$$

where $k_n^N(x;p) = (-N)_n K_n^N(x;p)$. The bivariate Krawtchouk polynomials are orthogonal with respect to the trinomial distribution function:

$$\sum_{0 \le x+y \le N} \binom{N}{x,y} p^x q^y (1-p-q)^{N-x-y} T_{i,j}(x,y) T_{k,l}(x,y) = \frac{(1-p-q)^{i+j}}{\binom{N}{i,j}\tilde{p}^i\tilde{q}^j} \delta_{i,k}\delta_{j,l},$$

where $\tilde{p} = \frac{p(1-p-q)}{1-p}, \tilde{q} = \frac{q}{1-p}$. These polynomials are also known to satisfy the 3-term recurrence relations involving multiplication by $x$

$$\begin{aligned} x T_{i,j}^N(x,y) = &-p(N-i-j)(T_{i+1,j}^N(x,y) - T_{i,j}^N(x,y)) \\ &-(1-p)i(T_{i-1,j}^N(x,y) - T_{i,j}^N(x,y)) \end{aligned} \tag{5}$$

and the 7-term recurrence relations when multiplied by $y$

$$\begin{aligned} y T_{i,j}^N(x,y) = &\frac{pq}{1-p}(N-i-j)(T_{i+1,j}^N(x,y) - T_{i,j}^N(x,y)) \\ &-\frac{q}{1-p}(N-i-j)(T_{i,j+1}^N(x,y) - T_{i,j}^N(x,y)) \\ &+qi(T_{i-1,j}^N(x,y) - T_{i,j}^N(x,y)) \\ &-(1-p-q)j(T_{i,j-1}^N(x,y) - T_{i,j}^N(x,y)) \\ &-\frac{p(1-p-q)}{1-p}j(T_{i+1,j-1}^N(x,y) - T_{i,j}^N(x,y)) \\ &-\frac{q}{1-p}i(T_{i-1,j+1}^N(x,y) - T_{i,j}^N(x,y)). \end{aligned} \tag{6}$$

Furthermore, one has the generating function formula [15]:

$$\sum_{0 \le x+y \le N} \binom{N}{x,y} s^x t^y T_{i,j}^N(x,y) = (1+s+t)^{N-i-j}(1+\frac{p-1}{p}s+t)^i(1+\frac{p+q-1}{q}t)^j. \tag{7}$$

In the following, set

$$p = \frac{1}{2}, \quad q = \frac{1}{4}$$

and introduce the orthonormal bivariate Krawtchouk polynomials

$$t_{i,j}^N(x,y) = \sqrt{\binom{N}{i,j}\tilde{p}^i\tilde{q}^j(1-p-q)^{-i-j}} T_{i,j}^N(x,y).$$

From (5) and (6), one can obtain for $\{t_{i,j}^N(x,y)\}$ the following contiguity relation:

$$\begin{aligned} \lambda_{x,y} t_{i,j}^N(x,y) = &\alpha\sqrt{(i+1)(N-i-j)}t_{i+1,j}^N(x,y) + \beta\sqrt{2(j+1)(N-i-j)}t_{i,j+1}^N(x,y) \\ &+\alpha j t_{i,j}^N(x,y) + \alpha\sqrt{i(N+1-i-j)}t_{i-1,j}^N(x,y) \\ &+\beta\sqrt{2j(N+1-i-j)}t_{i,j-1}^N(x,y) \\ &+\beta\sqrt{2i(j+1)}t_{i-1,j+1}^N(x,y) + \beta\sqrt{2(i+1)j}t_{i+1,j-1}^N(x,y), \end{aligned} \tag{8}$$

where the spectrum $\lambda_{x,y}$ is given by

$$\lambda_{x,y} = \alpha(N - 2x) + \beta(2N - 2x - 4y). \tag{9}$$

It is a straightforward matter to identify the correspondance between the projection (4) to the spin lattice of the quantum walk on the ordered Hamming graph $G_{\alpha,\beta}$ and the above recurrence relations for bivariate Krawtchouk polynomials (8).

## 5 Transfer properties on the graphs

Let us now examine the properties of the quantum walk on the ordered Hamming graph of depth 2 and of the projected dynamics on the spin lattice. With the motion initiated at $\lvert e_{0,0})$, the essential quantity is the transition amplitude

$$f_{(i,j)}(t) = \left(e_{i,j} \lvert e^{-itH} \rvert e_{0,0}\right).$$

From the correspondence between (4) and (8), the Hamiltonian (3) on 1-excitation subspace spanned by $\lvert e_{i,j})$ can be diagonalized by the bivariate Krawtchouk polynomials and its spectrum is given by (9). With the overlaps between the 1-excitation eigenstates of $H$ and the occupation basis states given by the orthonormalized polynomials $t_{i,j}^N(x,y)$ and using the generating function formula (7), one finds

$$f_{(i,j)}(t) = \sum_{0 \le x+y \le N} \binom{N}{x,y} \left(\frac{1}{2}\right)^x \left(\frac{1}{4}\right)^y \left(\frac{1}{4}\right)^{N-x-y} t_{0,0}^N(x,y) t_{(i,j)}^N(x,y) e^{-i\lambda_{x,y}t}$$

$$= e^{-iN(\alpha+2\beta)t} \frac{\sqrt{2^j}}{4^N} \sqrt{\binom{N}{i,j}} (1 + 2z_1 + z_2)^{N-i-j} (1 - 2z_1 + z_2)^i (1 - z_2)^j,$$

where $z_1 = e^{2(\alpha+\beta)ti}, z_2 = e^{4\beta ti}$. In [25, 28], fractional revival from the apex $(0,0)$ to the hypotenuse line $(i,j)$ $(i+j=N)$ was found in 2-dimensional $XX$-spin lattices related to the bivariate Krawtchouk polynomials of the Rahman type [12, 15, 18, 19]. To realize here a transfer to the same set or subset of points with $i + j = N$, it is easy to see that we should require that there be a time $t = T$ for which

$$1 + 2z_1 + z_2 = 0 \quad (\exists T \in \mathbb{R}). \tag{10}$$

Since $\lvert z_1 \rvert = \lvert z_2 \rvert = 1$, the relation (10) simultaneously imposes that

$$z_2 = 1 \tag{11}$$

at the same time $T$. Quite interestingly, these instances are the conditions for perfect state transfer:

$$\lvert f_{(N,0)}(T) \rvert = 1, \quad \lvert f_{(i,j)}(T) \rvert = 0 \quad ((i,j) \ne (N,0)).$$

Let us now clarify this. We can rewrite the condition (10) and (11) as follows:

$$e^{2i(\alpha+\beta)T} = -1, \quad e^{4i\beta T} = 1,$$

from where one finds

$$(2\alpha T, 2\beta T) = ((2m+1)\pi, 2n\pi), (2m\pi, (2n+1)\pi) \quad (m, n \in \mathbb{Z}).$$

Therefore, we can conclude that if

$$\frac{\alpha}{\beta} = \frac{(\text{even integer })}{(\text{odd integer})}$$

or

$$\frac{\alpha}{\beta} = \frac{(\text{odd integer})}{(\text{even integer})},$$

PST from $(0,0)$ to $(N,0)$ takes place at some time $T$.

The Fig.1 and Fig.2 are the plots of the transition probabilities of the graph $G_{\alpha,\beta}$ associated with $\alpha A_{(1,0)} + \beta A_{(0,1)}$.

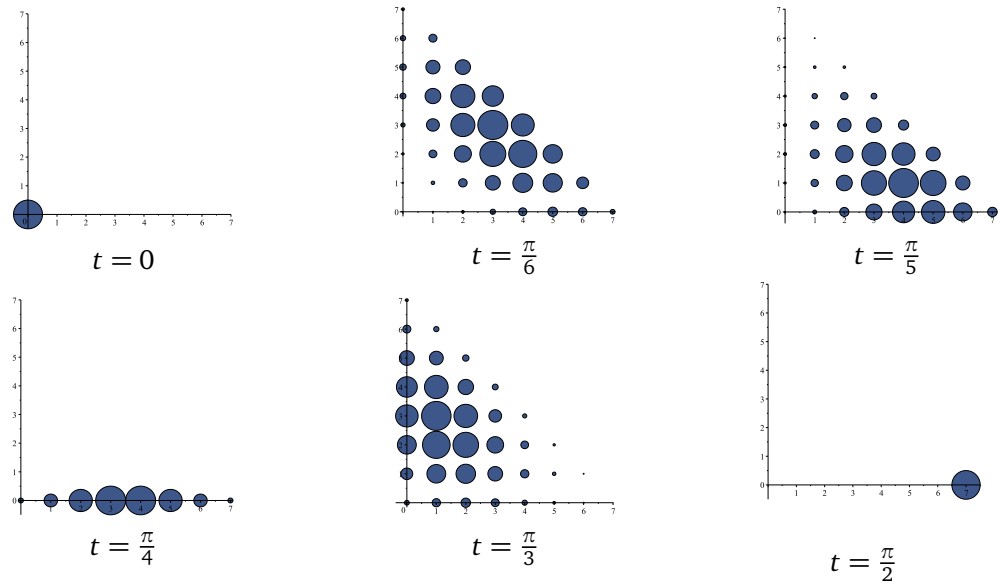

Figure 1: The transition amplitude $|f_{i,j}(t)|$ for $A_{(1,0)} + 2A_{(0,1)}$ when $N = 7$. The areas of the circles are proportional to $|f_{(i,j)}(t)|$ at the given lattice point $(i,j)$. PST occurs at $\frac{\pi}{2}$ and FR on the set of sites $i = 0, 1, \cdots, N$ and $j = 0$ occurs at $t = \frac{\pi}{4}$.

It should be noted here that PST also occurs on the graph $G_{0,1}$, whose projected lattice is of the shape given in Fig 3 and that the graph coincides with one in [14] when $N = 2, 3$. On all these graphs, PST occurs from $(0,0)$ to the farthest point $(N,0)$, which is desirable for quantum communication.

It was remarked that when $\beta = \frac{\alpha}{\sqrt{2}}$, the hopping terms in the Hamiltonian (4) are symmetric under rotations by $\frac{2}{3}\pi$. The spin network then identifies with the weight lattice of the fully symmetrized tensor product of the fundamental representation of $SU(3)$. That the bivariate Krawtchouk polynomials have an algebraic interpretation based upon $SU(3)$ has been established in [20] (see also [15]). For this specific choice of parameters ($\beta = \frac{\alpha}{\sqrt{2}}$), interestingly it is found that there is FR between the site $(0,N)$ and the lattice points $(i,0)$ $(i = 0, 1, \ldots, N)$. Indeed, for the transition amplitude

$$g_{(i,j)}(t) = \left( e_{i,j} | e^{-itH} | e_{0,N} \right),$$

there exists some time $T$ such that

$$\sum_{i=0}^{N} |g_{(i,0)}(T)|^2 = 1.$$

This is illustrated in Fig. 4.

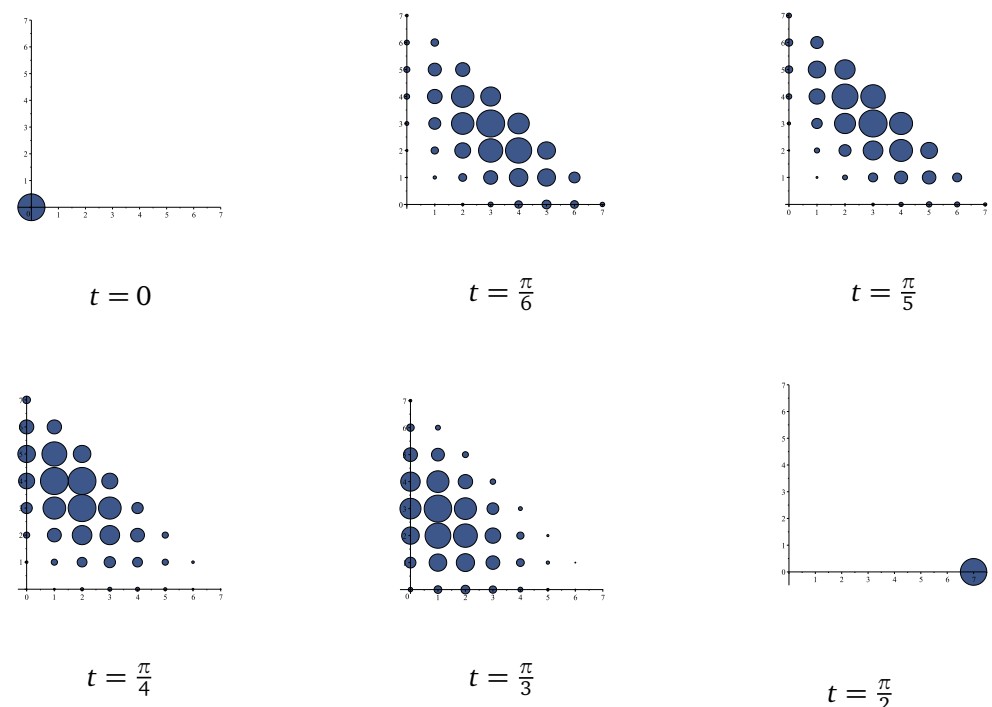

Figure 2: The transition amplitude $|f_{i,j}(t)|$ for $2A_{(1,0)} + A_{(0,1)}$ when $N = 7$. The areas of the circles are proportional to $|f_{(i,j)}(t)|$ at the given lattice point $(i, j)$. PST occurs at $\frac{\pi}{2}$.

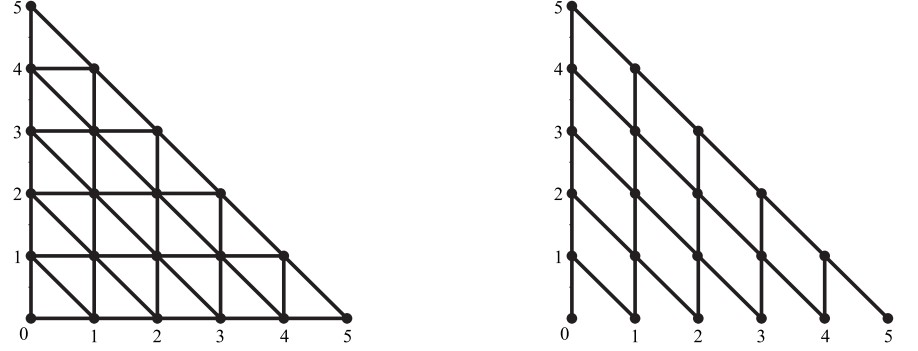

Figure 3: The projected graphs associated with $\alpha A_{(1,0)} + \beta A_{(0,1)}$ (left) and $A_{(0,1)}$ (right) when $N = 5$. On the graph associated with $A_{(0,1)}$, PST from $(0,0)$ to $(N,0)$ occurs at some time $T$.

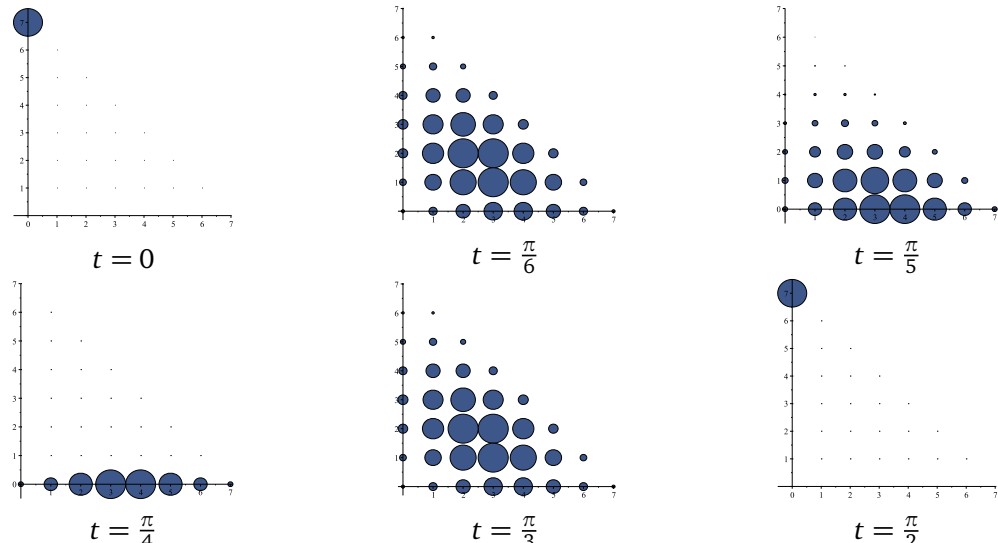

Figure 4: The transition amplitude $|g_{i,j}(t)|$ for $\sqrt{2}A_{(1,0)} + A_{(0,1)}$ when $N = 7$. The areas of the circles are proportional to $|g_{(i,j)}(t)|$ at the given lattice point $(i, j)$. FR on the set of sites $i = 0, 1, \cdots, N$ and $j = 0$ occurs at $\frac{\pi}{4}$.

## 6 Concluding Remarks

This paper has established the connection between quantum walks on graphs of the ordered Hamming scheme of depth 2 and the single excitation dynamics of certain two-dimensional lattices of triangular shape. This relation has featured the bivariate Krawtchouk polynomials of the Tratnik type that appear as eigenvalue matrices of the scheme and whose recurrence coefficients provide the couplings and Zeeman terms. We have focused on Hamiltonians $\alpha A_{(1,0)} + \beta A_{(0,1)}$ given by weighted combinations of the adjacency matrices of the two graphs associated to the shapes $(1, 0)$ and $(0, 1)$. Remarkably, when $\frac{\alpha}{\beta}$ is some rational number, we have observed that PST takes place between the sites $(0, 0)$ and $(N, 0)$ of the lattice at time $t = \frac{\pi}{2}$ after mixing on the whole two-dimensional lattice. In some examples, it has also been found that fractional revival occurs at $t = \frac{\pi}{4}$ at each of the sites of one side only of the lattice.

It should be stressed that the spin lattice that has been found here differs from the one discussed in [25] which is based on the more general Krawtchouk polynomials of Griffiths [12, 15, 18, 19]. The question of determining the graph to which the model in [25] lifts thus remains. The results presented here enrich the catalog of pairings between quantum walks on graphs and spin models in the context of PST. It is likely that PST could be preserved in the higher spin simplices related to graphs of the ordered Hamming scheme of depth $r$ where the multivariate Krawtchouk polynomials will intervene. It would prove interesting if such coherent transport could be realized in photonic lattices (see for instance [11, 27]). Finally, we would like to examine if the peculiar transport properties of the spin lattices could be of use in the design of certain algorithms.

## Acknowledgements

The authors would like to thank Matthias Christandl for asking about lifts to graphs of coherent transport on spin lattices. They are grateful to Paul Terwilliger for bringing references [3] and [26] to their attention. They also thank Ryo Sato and Kengo Miura for discussions. The

insightful inputs from Kareljan Schoutens and William Martin has also been much appreciated. LV wishes to acknowledge the hospitality of Kyoto University where most of this research was carried out. The research of ST is supported by JSPS KAKENHI (Grant Numbers 16K13761) and that of LV by a discovery grant of the Natural Sciences and Engineering Research Council (NSERC) of Canada.

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
