# Peer review of "Quantum Walks on Graphs of the Ordered Hamming Scheme and Spin Networks"

_SciPost Physics, doi:SciPost Phys. 7, 001 (2019)_

## Round 4 · Referee Report · Kareljan Schoutens (Referee 1) · 2019-5-13

Strengths
- Interesting results for quantum walks on 2D lattices, including novel examples of Perfect State Transfer (PST)
- Geometric origin of these walks through projection of walks on ordered 2-Hamming scheme
Weaknesses
None
Report
Quantum walks on the Krawtchouk chain (with non-uniform couplings) famously exhibit PST between the ends of the chain. It has long been understood that such walks can be understood by projecting walks on a simple hypercube (with uniform couplings). The present paper generalizes this observation to walks on 2D lattices that can be understood as projections from a higher-D space known as a ordered 2-Hamming scheme.
The treatment in the paper is very complete. It provides all necessary details on the ordered 2-Hamming scheme, the projection to the spin lattice, the eigenstates in terms of bivariate Krawtchouk polynomials and the dynamics leading to PST.
I have the following observation. Putting $\beta=\alpha/\sqrt{2}$, the hopping terms in the hamiltonian (4) acquire a symmetry over 12o degree rotations. In fact, both the hopping terms and the diagonal terms can be viewed as generators of an SU(3) symmetry, acting on the weight lattice of the representation which is the fully symmetrized tensor product of $N$ copies of the fundamental 3-dimensional representation of the Lie algebra SU(3). This is similar to the interpretation of the Krawtchouk chains as weight spaces of an SU(2) symmetry. For this choice of parameters, the quantum walks include special cases of interest, such as a PST from the corner $(0,N)$ to the base points $(i,0)$, $0\leq i \leq N$. It would be interesting to see how this symmetry is expressed in the algebraic setting of the bivariate Krawtchouk polynomials and in the geometric setting of the `covering' walk on the ordered 2-Hamming scheme.
The treatment in the paper is very complete. It provides all necessary details on the ordered 2-Hamming scheme, the projection to the spin lattice, the eigenstates in terms of bivariate Krawtchouk polynomials and the dynamics leading to PST.
I have the following observation. Putting $\beta=\alpha/\sqrt{2}$, the hopping terms in the hamiltonian (4) acquire a symmetry over 12o degree rotations. In fact, both the hopping terms and the diagonal terms can be viewed as generators of an SU(3) symmetry, acting on the weight lattice of the representation which is the fully symmetrized tensor product of $N$ copies of the fundamental 3-dimensional representation of the Lie algebra SU(3). This is similar to the interpretation of the Krawtchouk chains as weight spaces of an SU(2) symmetry. For this choice of parameters, the quantum walks include special cases of interest, such as a PST from the corner $(0,N)$ to the base points $(i,0)$, $0\leq i \leq N$. It would be interesting to see how this symmetry is expressed in the algebraic setting of the bivariate Krawtchouk polynomials and in the geometric setting of the `covering' walk on the ordered 2-Hamming scheme.
Requested changes
- section 2: a picture of a simple (low-$N$) case of the 2-ordered Hamming scheme would be helpful.
- typo on page 4: the number of columns in V is $\binom{N+2}{2}$, not $\binom{N+1}{2}$.
- I invite the authors to comment on the special choice $\beta=\alpha/\sqrt{2}$ where the walks have an SU(3) symmetry and where corner-to-base PST arises.

---

## Round 4 · Referee Report · William Martin (Referee 2) · 2019-5-14

Report
I have reviewed the paper "Quantum Walks on Graphs of the Ordered 2-Hamming Scheme and Spin Networks” by H. Miki, S. Tsujimoto and L. Vinet.
I apologize that time constraints prevent me from including more detail here.
The paper considers a certain weighted graph in the binary ordered Hamming scheme of depth two and shows that it exhibits perfect state transfer
and fractional revival. These phenomena are rare.
I recommend the paper for publication.
When Martin and Stinson discovered the ordered Hamming scheme, they made a few mistakes in their rush to publish. First, although they had
their own proofs, they noticed a preprint of Godsil which covered some of the material. In deference to Godsil, they deleted their own proofs and
gave credit to Godsil. Then Godsil never published that paper! They also used generating functions to find the eigenvalues — this became
cumbersome. Jürgen Bierbrauer was first to encode all of this in these multivariate Krawtchouk polynomials. Martin showed this material to
Alexander Barg, who knows coding theory better than anyone. All of this was unified, rewritten, and summarized in his student’s thesis
(Purkayastha) and the paper of Barg & Purkayastha.
Every association scheme can be viewed as a commutative matrix algebra. But there is an interesting non-commutative extension called the
Terwilliger algebra (or subconstituent algebra). For the binary Hamming scheme, this is a representation of the Lie algebra $sl_2(C)$. In unpublished
work (really guided by Terwilliger), Martin showed that the binary ordered Hamming scheme with depth r (which Vinet, et al. would call the ordered
r-Hamming scheme, notation that this reviewer sees as unwise) admits an action of the Lie algebra $sl_{r+1}(C)$ on its standard module where the
representation is again the Terwillger algebra. So it is correct — and very nice! — that one sees $SU(3)$ showing up here. This phenomenon likely
continues for larger values of depth r.
Perfect state transfer is incredibly rare and the binary Hamming scheme is the prototypical example where it occurs. It is interesting, therefore to
see the case r=2 also giving us perfect state transfer. I think the paper is very nice and well written. I do not like the terminology “ordered r-Hamming
scheme”, mostly because, for r=2, it becomes unclear which parameter is set to two. The only precedent I see in the literature is Bierbrauer’s
“ordered Hamming scheme of depth r”, which I would prefer over the hyphenation. Since Bierbrauer was first to formulate the eigenvalues as
evaluations of multivariate Krawtchouk polynomials, his paper should be included among the references:
http://citeseerx.ist.psu.edu/viewdoc/download?doi=10.1.1.73.4946&rep=rep1&type=pdf
Two very minor suggestions:
Page 2, line -2: “transport”? Or “transfer”?
Page 5, line -5 and Page 7, line 6 of section 5: I am unfamiliar with the notation $| x )$ and $( x | U | y )$ [instead of $|x>$, etc.]. Perhaps a word of explanation may help the reader coming from mathematics.
I apologize that time constraints prevent me from including more detail here.
The paper considers a certain weighted graph in the binary ordered Hamming scheme of depth two and shows that it exhibits perfect state transfer
and fractional revival. These phenomena are rare.
I recommend the paper for publication.
When Martin and Stinson discovered the ordered Hamming scheme, they made a few mistakes in their rush to publish. First, although they had
their own proofs, they noticed a preprint of Godsil which covered some of the material. In deference to Godsil, they deleted their own proofs and
gave credit to Godsil. Then Godsil never published that paper! They also used generating functions to find the eigenvalues — this became
cumbersome. Jürgen Bierbrauer was first to encode all of this in these multivariate Krawtchouk polynomials. Martin showed this material to
Alexander Barg, who knows coding theory better than anyone. All of this was unified, rewritten, and summarized in his student’s thesis
(Purkayastha) and the paper of Barg & Purkayastha.
Every association scheme can be viewed as a commutative matrix algebra. But there is an interesting non-commutative extension called the
Terwilliger algebra (or subconstituent algebra). For the binary Hamming scheme, this is a representation of the Lie algebra $sl_2(C)$. In unpublished
work (really guided by Terwilliger), Martin showed that the binary ordered Hamming scheme with depth r (which Vinet, et al. would call the ordered
r-Hamming scheme, notation that this reviewer sees as unwise) admits an action of the Lie algebra $sl_{r+1}(C)$ on its standard module where the
representation is again the Terwillger algebra. So it is correct — and very nice! — that one sees $SU(3)$ showing up here. This phenomenon likely
continues for larger values of depth r.
Perfect state transfer is incredibly rare and the binary Hamming scheme is the prototypical example where it occurs. It is interesting, therefore to
see the case r=2 also giving us perfect state transfer. I think the paper is very nice and well written. I do not like the terminology “ordered r-Hamming
scheme”, mostly because, for r=2, it becomes unclear which parameter is set to two. The only precedent I see in the literature is Bierbrauer’s
“ordered Hamming scheme of depth r”, which I would prefer over the hyphenation. Since Bierbrauer was first to formulate the eigenvalues as
evaluations of multivariate Krawtchouk polynomials, his paper should be included among the references:
http://citeseerx.ist.psu.edu/viewdoc/download?doi=10.1.1.73.4946&rep=rep1&type=pdf
Two very minor suggestions:
Page 2, line -2: “transport”? Or “transfer”?
Page 5, line -5 and Page 7, line 6 of section 5: I am unfamiliar with the notation $| x )$ and $( x | U | y )$ [instead of $|x>$, etc.]. Perhaps a word of explanation may help the reader coming from mathematics.

---

## Round 5 · Referee Report · Kareljan Schoutens (Referee 1) · 2019-6-19

Report
I am satisfied with the changes made to the manuscript and recommend that it be published in SciPost.

---

## Round 5 · Author Response

We are grateful to both referees for their valuable comments. Their deep insight help to improve the contents and presentation of our manuscript.
Following their comments, we made several changes and some additions in the manuscript.
-response to report 1:
We fixed typos and added the case alpha=beta/sqrt{2} case which corresponds to the model with SU(3) symmetry.
As pointed out, the FR from corner to base points actually takes place and we gave the corresponding figure.
The 2-variable Krawtchouk polynomials of course have an algebraic interpretation based upon this symmetry and we added the corresponding references.
We tried to put simple figures about ordered Hamming scheme of depth 2. However, for N=1, there are 3 graphs G_{0,0} (4 nodes without edge), G_{0,1} (square), G_{1,0} (4 nodes with 2 edges) and they are too simple. For N=2, we have 6 graphs (each has 16 nodes and several edges) and they are too complicated for this manuscript. Therefore, we did not put the figures of the ordered Hamming graph in the manuscript.
-response to report 2:
We agree with suggestions and changed the terminology from ordered r-Hamming scheme to ordered Hamming scheme of depth r in the manuscript including its title.
We changed the description about |e_{i,j}) from (N+1)x(N+1) matrix to orthonormal basis vector which might be easier to understand.
We also added the references and fixed some typos.
Following their comments, we made several changes and some additions in the manuscript.
-response to report 1:
We fixed typos and added the case alpha=beta/sqrt{2} case which corresponds to the model with SU(3) symmetry.
As pointed out, the FR from corner to base points actually takes place and we gave the corresponding figure.
The 2-variable Krawtchouk polynomials of course have an algebraic interpretation based upon this symmetry and we added the corresponding references.
We tried to put simple figures about ordered Hamming scheme of depth 2. However, for N=1, there are 3 graphs G_{0,0} (4 nodes without edge), G_{0,1} (square), G_{1,0} (4 nodes with 2 edges) and they are too simple. For N=2, we have 6 graphs (each has 16 nodes and several edges) and they are too complicated for this manuscript. Therefore, we did not put the figures of the ordered Hamming graph in the manuscript.
-response to report 2:
We agree with suggestions and changed the terminology from ordered r-Hamming scheme to ordered Hamming scheme of depth r in the manuscript including its title.
We changed the description about |e_{i,j}) from (N+1)x(N+1) matrix to orthonormal basis vector which might be easier to understand.
We also added the references and fixed some typos.

---

## Round 5 · List of Changes

-We corrected some typos.
-We added references [5],[14] and [20].
-We changed the terminology from ordered r-Hamming scheme to ordered Hamming scheme of depth r. We then changed the title according to it.
-We added case alpha=beta/sqrt{2} in p. 7. Some description and one figure about this case were given.
-We added references [5],[14] and [20].
-We changed the terminology from ordered r-Hamming scheme to ordered Hamming scheme of depth r. We then changed the title according to it.
-We added case alpha=beta/sqrt{2} in p. 7. Some description and one figure about this case were given.

---

## Editorial Decision

published